# Regulation of Tumor Microenvironment through YAP/TAZ under Tumor Hypoxia

**DOI:** 10.3390/cancers16173030

**Published:** 2024-08-30

**Authors:** Sung Hoon Choi, Do Young Kim

**Affiliations:** 1Institute of Health & Environment, Graduate School of Public Health, Seoul National University, Seoul 08826, Republic of Korea; burnes@kobiolabs.com; 2KoBioLabs Inc., Seoul 08826, Republic of Korea; 3Department of Internal Medicine, Yonsei University College of Medicine, 50-1 Yonsei-ro, Seodaemun-gu, Seoul 03722, Republic of Korea; 4Yonsei Liver Cancer Center, Yonsei Cancer Hospital, Seoul 03722, Republic of Korea

**Keywords:** tumor hypoxia, YAP, TAZ, HIF, Hippo pathway

## Abstract

**Simple Summary:**

The YAP/TAZ signaling pathways, which is involved in tumor development and proliferation in a variety of solid tumors, inhibits hypoxia-induced cell death, increases and stabilizes angiogenesis, and activates tumor proliferation, both independently and in conjunction with HIF in hypoxic tumors. It also promotes tumor metastasis by regulating different TME environments. Therefore, understanding the activity of YAP/TAZ as a mechanism of tumor development may lead to the development of novel therapeutic strategies.

**Abstract:**

In solid tumors such as hepatocellular carcinoma (HCC), hypoxia is one of the important mechanisms of cancer development that closely influences cancer development, survival, and metastasis. The development of treatments for cancer was temporarily revolutionized by immunotherapy but continues to be constrained by limited response rates and the resistance and high costs required for the development of new and innovative strategies. In particular, solid tumors, including HCC, a multi-vascular tumor type, are sensitive to hypoxia and generate many blood vessels for metastasis and development, making it difficult to treat HCC, not only with immunotherapy but also with drugs targeting blood vessels. Therefore, in order to develop a treatment strategy for hypoxic tumors, various mechanisms must be explored and analyzed to treat these impregnable solid tumors. To date, tumor growth mechanisms linked to hypoxia are known to be complex and coexist with various signal pathways, but recently, mechanisms related to the Hippo signal pathway are emerging. Interestingly, Hippo YAP/TAZ, which appear during early tumor and normal tumor growth, and YAP/TAZ, which appear during hypoxia, help tumor growth and proliferation in different directions. Peculiarly, YAP/TAZ, which have different phosphorylation directions in the hypoxic environment of tumors, are involved in cancer proliferation and metastasis in various carcinomas, including HCC. Analyzing the mechanisms that regulate the function and expression of YAP in addition to HIF in the complex hypoxic environment of tumors may lead to a variety of anti-cancer strategies and combining HIF and YAP/TAZ may develop the potential to change the landscape of cancer treatment.

## 1. Introduction

In many solid cancers, such as hepatocellular carcinoma (HCC), tumor survival and development is maintained by multiple complex, dense, and overlapping signaling pathways that allow the cancer to proliferate, develop, and metastasize [1]. In addition, the treatment of advanced solid tumors through immunosuppression, angiogenesis, etc., is challenging due to this complex and diverse signaling system [2]. In particular, hypoxia and hypoxia-inducible factor (HIF) signaling, which is known to act as a tumorigenic angiogenesis switch, promote cancer progression and treatment resistance [3] and have been reported in solid cancers such as hepatocellular carcinoma (HCC) [4].

The Hippo pathway is a signaling pathway that plays a crucial role in regulating cell proliferation, apoptosis, and organ size [5]. Dysregulation of the Hippo pathway has been implicated in various cancers, including liver cancer [5]. Tumor hypoxia, on the other hand, refers to the condition where the tumor microenvironment experiences low levels of oxygen due to rapid tumor growth and insufficient blood supply [4].

In liver cancer, there is evidence to suggest a correlation between the Hippo pathway and tumor hypoxia [6]. One of the key components of the Hippo pathway is Yes-associated protein (YAP) and its corresponding transcriptional activator with a PDZ-binding motif (TAZ) [7]. When the Hippo pathway is activated, YAP/TAZ activity are inhibited, resulting in inhibition of cell proliferation and promotion of apoptosis [6]. However, when the Hippo pathway is disrupted, YAP/TAZ become active and translocate to the nucleus, where they interact with transcription factors to promote cell proliferation and survival [7,8].

Tumor hypoxia, on the other hand, triggers various cellular responses, including activation of HIFs [9]. HIFs are transcription factors that regulate the expression of genes involved in adaptation to low oxygen conditions, such as angiogenesis, glycolysis, and cell survival [4,9,10].

Studies have shown the presence of crosstalk between the Hippo pathway and HIF signaling under hypoxic conditions. For instance, YAP has been reported to interact with HIF-1α, a subunit of HIF, and regulate its transcriptional activity [11]. Additionally, YAP has been shown to promote the expression of HIF-1α target genes involved in glycolysis and angiogenesis, thereby contributing to tumor growth and progression under hypoxic conditions [11,12].

Therefore, the dysregulation of the Hippo pathway, leading to increased YAP/TAZ activity, can promote tumor growth and survival under hypoxic conditions in liver cancer. Understanding the interplay between the Hippo pathway and tumor hypoxia may provide insights into the development of novel therapeutic strategies for liver cancer.

## 2. The Protection of DNA Damage by YAP/TAZ under Hypoxia

The protection of DNA damage by YAP/TAZ under hypoxia is an intriguing aspect of cellular adaptation to low oxygen conditions, particularly in the context of cancer progression [11]. YAP/TAZ, regulatory transcriptional co-activators of the DNA repair pathway, promote the transcription of genes involved in DNA repair mechanisms under tumor hypoxic conditions. [13,14], such as nucleotide excision repair (NER), base excision repair (BER), and homologous recombination (HR). By upregulating these DNA repair pathways, YAP/TAZ help maintain genomic stability and protect cells from accumulating DNA damage [14,15].

(1)Inhibition of apoptosisYAP/TAZ activation under hypoxia can also suppress apoptosis, allowing cells to survive despite DNA damage [14]. HIFs, which are activated under low oxygen conditions, can interact with YAP/TAZ to regulate the expression of anti-apoptotic genes [6]. By inhibiting apoptosis, YAP/TAZ contribute to cell survival, even in the presence of DNA damage, thereby protecting cells from undergoing programmed cell death [6,13,15]. For example, YAP can trigger apoptosis by binding p73 instead of TEAD, thereby upregulating the anti-apoptotic gene [16]. In addition, inhibition of YAP signaling can promote apoptosis in multiple pathways. Knockdown of YAP and TAZ can enhance apoptosis under hypoxic condition [17].(2)Promotion of cell cycle progressionYAP/TAZ activation under hypoxia can promote cell cycle progression, facilitating the proliferation of damaged cells [17]. This effect is mediated through the transcriptional regulation of cell cycle-related genes by YAP/TAZ [18]. By promoting cell cycle progression, YAP/TAZ contribute to the proliferation of cells with DNA damage, potentially leading to tumor progression and expansion despite the presence of hypoxia-induced genotoxic stress [14,17]. YAP activation was increased, and this facilitated cell cycle progression through RhoA and cytoskeletal dynamics. Increased YAP and TEAD activity lead to marked expansion of the neural progenitor population by facilitating cell cycle progression through induction of cyclin and cyclin dependent kinase [19,20].(3)Development of tumor angiogenesisTumor angiogenesis is activated by endothelial proliferation, collective cell migration, and cellular rearrangements under tumor hypoxia [21]. Angiogenic stimuli, such as VEGF and FGF, activate endothelial cells (ECs) to promote the formation of endothelial tip cells that migrate toward the proliferating tumor [22]. During development, the formation of new blood vessels is accompanied by a decrease in angiogenic growth factor levels, and the vessels mature through stabilized cell–cell junctions and recruiting endothelial cells to the walls of the vessels. In mature vessels, the induction of ECs is arrested and YAP/TAZ are inactivated in the ECs [22,23]. During wound healing, YAP/TAZ are activated as needed to induce angiogenesis in various inflammatory and wound conditions [7]. However, in the tumor microenvironment, the angiogenic process is to some extent similar to that of inflamed wound tissue, and, especially in the hypoxic phase [23], YAP/TAZ remain activated in the endothelial cells and the vasculature does not develop to a mature state; instead, immature and incomplete vessels are formed [24]. This cycle is repeated every time the tumor proliferates, and YAP/TAZ activity in the ECs is also maintained in an active phase without an inactive phase [7,21,22]. The resulting immature vessels repeatedly induce tumor proliferation and metastasis to other organs or tissues [25]. The development of these vessels is prominent in multivessel tumors such as liver cancer, and the activity of YAP/TAZ in the ECs is also affected by the hypoxic tumor microenvironment.(4)Interaction with other signaling pathwaysYAP/TAZ can interact with various signaling pathways involved in DNA damage response and repair, such as the p53 pathway and the ATM/ATR kinase pathway [26]. Under hypoxic conditions, YAP/TAZ may modulate the activity of these pathways to promote cell survival and DNA repair [27]. Additionally, YAP/TAZ can crosstalk with other hypoxia-responsive transcription factors, such as HIFs, to coordinate cellular responses to low oxygen levels and genotoxic stress [26,27]. And in phosphorylation-independent pathways, such as the Wnt and hormone signaling pathways, mechanical signals are transmitted to the nucleus via stress fibers and actin remodeling. Unphosphorylated YAP/TAZ can undergo nuclear translocation via Rho or beta-catenin [28].

Overall, the protection of DNA damage by YAP/TAZ under hypoxia involves the regulation of DNA repair pathways, inhibition of apoptosis, promotion of cell cycle progression, angiogenesis, and crosstalk with other signaling pathways, such as p53, ATM, Wnt, etc. [13,27]. Understanding these mechanisms may provide insights into the development of novel therapeutic strategies targeting YAP/TAZ signaling in cancer cells exposed to hypoxic microenvironments (Table 1).

## 3. HIF-1α Interacts with YAP and Promotes Nuclear Translocation

HIP-1alpha (HIF-1α) is a key transcription factor that plays a central role in cellular responses to low oxygen levels [4]. It regulates the expression of genes involved in various biological processes, including angiogenesis, glycolysis, cell survival, and metastasis [4,10]. YAP is a transcriptional coactivator that is a central component of the Hippo signaling pathway, which regulates organ size, tissue homeostasis, and tumorigenesis [6].

Under hypoxic conditions, HIF-1α protein stability increases due to reduced oxygen availability [10]. This stabilization prevents the degradation of HIF-1α, allowing it to translocate to the nucleus [49]. Once in the nucleus, HIF-1α forms a heterodimer with HIF-1β (also known as ARNT), and this complex binds to hypoxia-responsive elements (HREs) in the promoters of target genes, thereby activating their transcription [10,49].

Recent studies have indicated that HIF-1α can interact with YAP, forming a protein complex that promotes nuclear translocation [11]. This interaction between HIF-1α and YAP can occur through direct physical binding or through indirect mechanisms involving other proteins or signaling pathways.

Once formed, the HIF-1α/YAP complex translocates to the nucleus, where it can modulate the transcriptional activity of target genes [11,50]. In some cases, this complex may cooperate with other transcription factors or coactivators to enhance gene expression synergistically [51].

The nuclear translocation of the HIF-1α/YAP complex facilitates the activation of genes involved in promoting cell survival, adaptation to hypoxic stress, and tumorigenesis [5,26]. These genes include those encoding for glycolytic enzymes, angiogenic factors, and anti-apoptotic proteins, among others [18].

Overall, the interaction between HIF-1α and YAP, and their subsequent nuclear translocation, represent a mechanism by which cells respond to hypoxic stress and regulate gene expression to promote survival and adaptation to low oxygen environments [18]. This crosstalk between HIF-1α and YAP underscores the complexity of cellular responses to hypoxia and suggests potential targets for therapeutic intervention in diseases such as cancer, where hypoxia and dysregulated Hippo signaling contribute to tumor progression.

## 4. Differential Regulation of YAP and TAZ under Hypoxia

The regulation of YAP and its analogue, the transcriptional activator TAZ, under hypoxic conditions involves complex interactions with a variety of signaling pathways and transcription factors. [14,52]. Here are some key differences in the regulation of YAP and TAZ under hypoxia:(1)Regulatory Mechanisms:Under hypoxia, YAP can be regulated through both transcriptional and post-translational mechanisms. HIF-1α can directly interact with YAP, promoting its nuclear translocation and activation of target genes [11]. Additionally, hypoxia may affect the expression of upstream regulators of YAP, such as the Hippo pathway components MST1/2 and LATS1/2, leading to altered YAP activity [53,54]. On the other hand, TAZ regulation under hypoxia appears to involve similar mechanisms as YAP, including direct interaction with HIF-1α [55]. However, the specific regulatory pathways and the extent of TAZ activation under hypoxia may differ from YAP [55]. Additionally, TAZ may have distinct binding partners or post-translational modifications that influence its activity in response to hypoxic stress [56,57].(2)Transcriptional Targets:YAP can activate the expression of genes involved in promoting cell survival, proliferation, and angiogenesis, contributing to tumor growth and progression under hypoxia [7,17,52]. These target genes may include those encoding for angiogenic factors, glycolytic enzymes, and anti-apoptotic proteins [17]. TAZ shares many transcriptional targets with YAP and can similarly regulate genes involved in cell proliferation, survival, and tissue growth under hypoxic conditions [58]. However, TAZ may also have unique target genes or regulate gene expression in a context-dependent manner, leading to distinct cellular outcomes compared to YAP [59].(3)Cellular Localization:Under normoxic conditions, YAP is predominantly localized in the cytoplasm, where it undergoes phosphorylation-mediated inhibition by Hippo pathway kinases [59]. However, under hypoxia, YAP can translocate to the nucleus, where it interacts with transcription factors such as HIF-1α to regulate gene expression [11]. Similar to YAP, TAZ is regulated by phosphorylation and predominantly localized in the cytoplasm under normoxia conditions [56]. Upon hypoxic stimulation, TAZ can also translocate to the nucleus and participate in transcriptional regulation, potentially through interaction with HIF-1α or other nuclear factors [55].(4)Functional Roles:Not only YAP activation under hypoxia is associated with increased cell proliferation, survival, and angiogenesis, contributing to tumor growth and metastasis, but also TAZ activation under hypoxia likely plays a similar role in promoting cell survival, proliferation, and angiogenesis [52,55,56]. Although, its specific functions may vary depending on the cellular context and the repertoire of target genes regulated by TAZ [55,57,59].

Therefore, while both YAP and TAZ are regulated by hypoxic signaling pathways and contribute to cellular responses to low oxygen conditions [50], they may differ in their regulatory mechanisms, transcriptional target, cellular localization, and functional roles (Figure 1) [28,60]. Understanding these differences is crucial to unravelling the complex interplay between hypoxia signaling and the Hippo pathway in various physiological and pathological contexts.

## 5. YAP or TAZ Is Functionally Involved in Other Cancer Cells

TAZ is a transcriptional co-activator that plays a crucial role in various cellular processes, including cell proliferation, differentiation, migration, and stemness [61]. It is a key component of the Hippo signaling pathway, which regulates organ size, tissue homeostasis, and tumorigenesis [62,63]. Dysregulation of TAZ has been implicated in numerous types of cancer, making it a potential therapeutic target [63]. Here, is a detailed description of TAZ’s functional involvement in different types of cancer (Figure 2):

TAZ is overexpressed in triple-negative breast cancer (TNBC) and is associated with a poor prognosis. It promotes cancer cell proliferation, invasion, and metastasis by activating genes involved in epithelial–mesenchymal transition (EMT), such as Snail and Slug. TAZ also enhances cancer stem cell properties, contributing to tumor initiation and resistance to therapy. [64]. On the other hand, high YAP activity correlates with a high histological grade, enrichment of stem cell signatures, metastasis proclivity, and a poor outcome [65]. However, in TNBC, the effect of TAZ on tumor development is more significant than that of YAP.

In lung cancer, TAZ promotes tumor growth and metastasis by regulating genes involved in cell proliferation, apoptosis evasion, and EMT [66]. It interacts with various signaling pathways, including the Wnt/β-catenin pathway, to drive oncogenic processes [64,66]. TAZ expression correlates with advanced stages of lung cancer and is associated with reduced patient survival [66]. In the case of YAP, ectopic YAP expression can promote the progression of small adenomas to high-grade lung adenocarcinoma. In addition, overexpression of YAP can confer the ability to metastasize from benign breast tumors to the lungs [67,68].

TAZ is upregulated in prostate cancer (PC) and correlates with aggressive disease and poor prognosis [59,69]. It promotes cancer cell proliferation, invasion, and resistance to apoptosis by activating target genes involved in cell cycle progression and survival [56,59]. TAZ also enhances cancer stemness in prostate cancer, contributing to tumor recurrence and therapeutic resistance [63,69,70], whereas ectopic YAP expression triggers malignant transformation and increases tumor cell proliferation [71]. YAP induces PCa formation in an androgen independent manner, via promoting AKT and MEK–ERK pathway signaling [72]. YAP has a positive effect on the acquisition of an aggressive PCa cell phenotype and promotes cell motility and invasion [73].

In ovarian cancer, TAZ overexpression is associated with aggressive tumor behavior, chemotherapy resistance, and poor patient outcomes [74,75]. TAZ promotes cancer cell proliferation, migration, and invasion by activating genes involved in EMT, stemness, and drug resistance [64]. Targeting TAZ signaling has emerged as a potential therapeutic strategy to improve the outcomes of ovarian cancer patients [75,76]. Moreover, overexpression and activation of YAP induces increased proliferation, resistance to cisplatin-induced apoptosis, loss of contact inhibition, promotion of metastasis through increased cell motility, and anchorage-independent growth of ovarian cancer cells [77]. This promotion of ovarian cancer growth by YAP has been shown to be an indicator of poor prognosis in ovarian cancer patients [78].

TAZ is upregulated in colorectal cancer and is associated with advanced tumor stage, metastasis, and poor prognosis [79]. It enhances cancer cell proliferation, invasion, and metastasis by activating genes involved in EMT, stemness, and angiogenesis [64,80]. TAZ also crosstalks with other signaling pathways, such as the Notch and Wnt pathways, to promote colorectal cancer progression [74,80,81]. High expression levels of a gene signature for YAP activity have been shown to predict poor prognosis and correlate with resistance to cetuximab, one of the treatment strategies for colorectal cancer [82]. Thus, nuclear YAPs can interact with other transcription factors to promote cancer cell proliferation, apoptosis, metastasis, and stem cell maintenance [83].

Finally, in HCC, TAZ is frequently overexpressed and correlates with tumor progression and poor patient survival [84]. TAZ promotes HCC cell proliferation, migration, and invasion by activating genes involved in cell cycle regulation, EMT, and angiogenesis [61]. It also interacts with YAP (Yes-associated protein), another Hippo pathway effector, to synergistically drive HCC development and progression [61,74,84]. In addition, YAP is correlated with the stemness of liver cancer stem cells, and liver cancer stem cells are closely associated with YAP-induced tumor initiation and progression [85].

Overall, YAP and/or TAZ play a multifaceted role in different types of cancer, promoting tumor growth, metastasis, and therapeutic resistance [63]. Targeting YPA/TAZ signaling pathways holds promise for the development of novel cancer therapies aimed at inhibiting tumor progression and improving patient outcomes [58,62,63,64].

## 6. Hypoxic Conditions Had Opposing Roles in the Level of p-YAP and p-TAZ

Under hypoxic conditions, cells experience low oxygen levels, which can profoundly influence cellular signaling pathways and gene expression patterns [9]. The role of hypoxia in regulating the phosphorylation status of YAP and TAZ, two key effectors of the Hippo pathway, can vary depending on the context and cell type. [86]. Hypoxic conditions and their effects on the phosphorylation of YAP and TAZ were examined (Figure 3):

(1)YAP Phosphorylation: Hypoxia can lead to the stabilization and nuclear accumulation of YAP in some cellular contexts [87]. This is often mediated through the inactivation of the Hippo pathway, which normally phosphorylates YAP, leading to its cytoplasmic retention and degradation [7,88]. Under hypoxic conditions, decreased activity of the Hippo pathway kinases, such as LATS1/2 (Large Tumor Suppressor 1/2), may occur, resulting in reduced phosphorylation of YAP [89]. As a consequence, YAP is less likely to undergo degradation and more likely to translocate to the nucleus, where it acts as a transcriptional co-activator [90]. In certain cancer cells, hypoxia-induced YAP activation can promote cell survival, proliferation, and metastasis by regulating the expression of target genes involved in these processes [86,88,90]. YAP activation under hypoxia may thus contribute to tumor aggressiveness and therapy resistance [16,59].(2)TAZ Phosphorylation: Contrary to YAP, hypoxic conditions may lead to increased phosphorylation and cytoplasmic retention of TAZ in some cellular components [60]. This can occur through various mechanisms, including activation of the Hippo pathway or other signaling pathways that modulate TAZ phosphorylation. Hypoxia-induced TAZ phosphorylation may involve the activation of LATS1/2 or other kinases that directly phosphorylate TAZ, promoting its interaction with 14-3-3 proteins and sequestering it in the cytoplasm [91,92]. Cytoplasmic retention of phosphorylated TAZ under hypoxia prevents its nuclear translocation and transcriptional co-activation activity [44,93]. Consequently, the expression of TAZ target genes involved in cell proliferation, survival, and EMT may be downregulated [94]. In certain cellular contexts, hypoxia-induced inhibition of TAZ activity [19,95,96] may contribute to the suppression of tumorigenic processes, such as cell proliferation, invasion, and metastasis [94,97]. Therefore, the hypoxia phenomenon that occurs during tumor growth can itself affect tumor proliferation and metastasis through the HIF pathway but has a synergistic effect on the stabilization of tumor development and amplification of proliferation and metastasis by YAP pathway.

To summarize, hypoxic conditions can exert opposing effects on the phosphorylation levels and activities of YAP and TAZ, depending on the cellular components and the specific regulatory mechanisms involved [20,98]. While hypoxia-induced YAP activation may promote tumorigenesis and metastasis in certain components [99], hypoxia-induced TAZ inhibition may have tumor-suppressive effects by limiting the transcriptional activity of TAZ and its oncogenic functions [93,100]. Understanding the complex interplay between hypoxia and the Hippo pathway effectors YAP and TAZ is essential for elucidating their roles in cancer progression and identifying potential therapeutic targets.

## 7. Conclusions

Under hypoxic conditions, cells experience low levels of oxygen, which can profoundly affect cellular signaling pathways and gene expression patterns. YAP/TAZ are involved in tumor development and metastasis through several pathways under hypoxic conditions. YAP acts as a transcriptional activator by binding to HIF, and TAZ inhibits the process involved in tumor growth through phosphorylation; however, reducing the binding affinity with YAP promotes the binding of YAP to HIF, thereby promoting tumor growth. It can be seen that it affects development. Although many more studies still need to be conducted, the development of the YAP/TAZ signals, which happens through the HIPPO pathway, is greatly involved in tumor growth before activation of the HIF pathway, and when an HIF pathway component such as the angiogenic switch is activated, tumor growth caused by YAP/HIF is significantly involved. It can be said to affect both growth and metastasis. Therefore, if the activated/deactivated signaling pathway is utilized for the development of anticancer drugs or diagnostic agents, a better anticancer treatment strategy can be established.

## Figures and Tables

**Figure 1 cancers-16-03030-f001:**
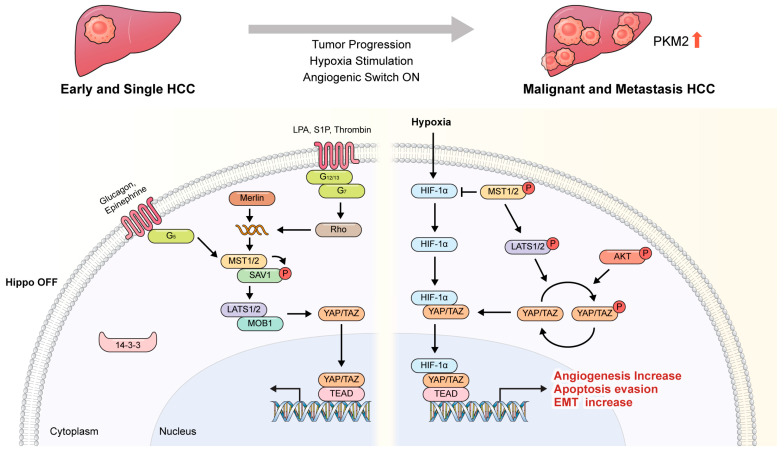
YAP/TAZ mechanisms in solid tumors, particularly HCCs, in hypoxic environments. In early tumor development, YAP/TAZ act as important cell signals, not only for tumor apoptosis, but also for tumor proliferation. However, when tumor growth develops excessively and a hypoxic environment is created, YAP separates from TAZ and enters blood vessels through HIF1a. By generating and activating EMT, it maintains the microenvironment around the tumor so that the tumor can develop better, and it also develops an environment for the tumor to metastasize to other organs or tissues. In early-stage tumors, YAP/TAZ bind to TEAD and act as a transcriptional activator through the Hippo off signaling pathway, following the G protein-coupled receptor (GPCR) or glucagon receptor (Gcgr) signaling system to induce the development of early-stage tumors. When a tumor grows beyond a certain size and is unable to obtain the oxygen and nutrients it needs to proliferate, the angiogenic switch is triggered and through the hypoxia signaling system, overexpressed HIF and YAP/TAZ combine with TEAD to activate the expression of various genes to overcome hypoxia. Specifically, when localized to the nucleus, YAP is recruited together with hypoxia-inducible factor 1α (HIF-1α) for PKM2 transcription at the pyruvate kinase M2 (*PKM2*) gene promoter, contributing to tumor formation and metastasis.

**Figure 2 cancers-16-03030-f002:**
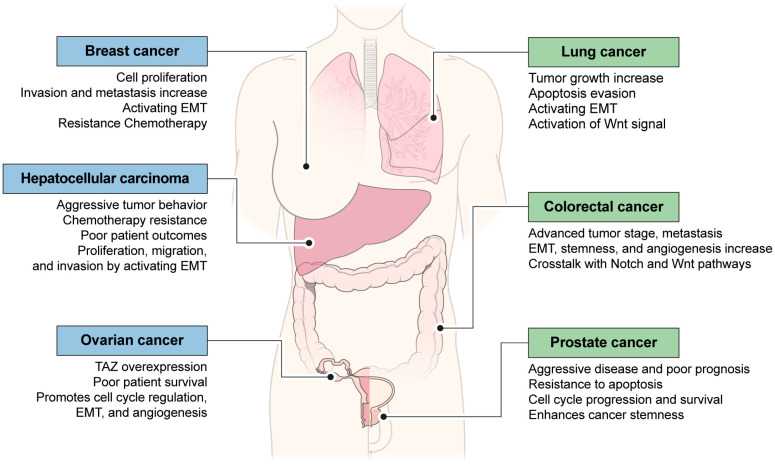
Tumor development through YAP/TAZ under hypoxic environment in various cancers. Changes in YAP/TAZ under hypoxia conditions are not unique to HCC. In various cancers, tumors develop through mechanisms that increase tumor proliferation and evade tumor cell death through YAP/TAZ. When a certain size is reached, the hypoxic stimulation and angiogenesis switch are activated in the tumor, and changes occur in the tumor microenvironment. YAP/TAZ also avoid cell death due to hypoxia through different phosphorylation mechanisms, leading to the development of metastasis and proliferation through increased EMT and resistance to chemotherapy. This is one of the main mechanisms of development in intractable tumors such as liver cancer, breast cancer, lung cancer, colon cancer, and ovarian cancer.

**Figure 3 cancers-16-03030-f003:**
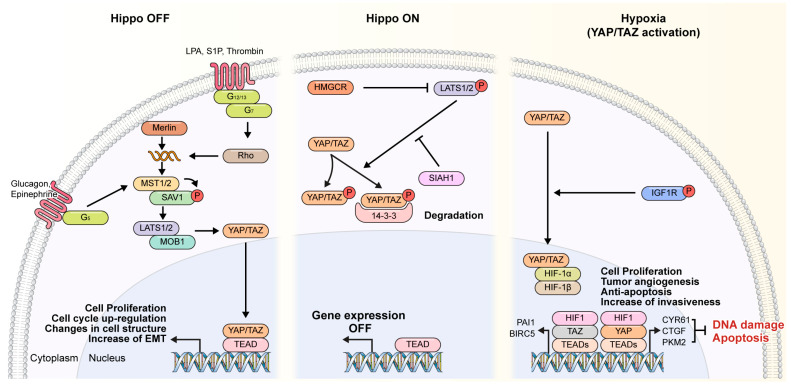
Phosphorylation of YAP/TAZ under hypoxic conditions. Under normoxia environments, phosphorylation of YAP/TAZ increases the expression of genes related to tumor growth through the combination of transcription activators; YAP/TAZ bind to TEAD and acts as a transcriptional activator through the Hippo-Off signaling pathway, mediating the translocation of the YAP/TEZ complex into the nucleus by inducing phosphorylation of Rho, MST1/2, and LAST1/2, in sequence, depending on the GPCR or GCGR signaling system. Once translocated into the nucleus, YAP/TAZ bind to TEAD and activate cell proliferation and the cell cycle, and changes in cellular structure, including hypertrophy of heterogeneous cells, leading to tumor development. However, under hypoxic environments, YAP and TAZ bind to HIF differently through a variant phosphorylation process. Hypoxia-mediated HIF-1a translocates from the cytosol to the nucleus, where it binds to unphosphorylated YAP and binds to TEAD. YAP/HIF binds to DNA, inhibits DNA damage and apoptosis, and induces the expression of a variety of genes involved in cell proliferation, angiogenesis, and metastasis. In addition, interaction of HIF-1α with TAZ also stimulates TAZ/TEAD transcriptional activity. TAZ and HIF-1α interact and function as mutual transcriptional cofactors. HIF-1α acts as a cofactor of the TAZ/TEADs complex for the transcription of target genes, and TAZ acts as a cofactor of HIF-1α for the transcription of target genes such as PAI1, BIRC5, CTGF, PDK1, and LDHA in the hypoxic tumor microenvironment. Moreover, TAZ also regulates the mechanism by which HIF binds to and regulates TAZ expression. These different hypoxic mechanisms of YAP/TAZ affect tumor formation and patient mortality and are involved in sensitivity to anticancer drugs and development of tumor metastasis.

**Table 1 cancers-16-03030-t001:** YAP/TAZ induce diverse tumor microenvironment functions.

Function	Gene	Major Pathway	References
**Anti-apoptosis**	Survivin	YAP promotes sorafenib resistance through upregulation of Survivin expression.	[29]
	CTGF	CTGF acts as a direct target gene for YAP, promoting cell proliferation and anchorage for independent growth. It also functions as a transcriptional co-repressor to promote cell survival by repressing DNA damage.	[30]
	AXL	AXL is a tyrosine kinase receptor that acts as the main downstream effector responsible for sustaining YAP-driven resistance. In addition, YAP and its downstream target AXL play a crucial role in resistance to EGFR TKIs.	[31]
	Bcl	Overexpression of TAZ may upregulate its target genes, including connective tissue growth factor (CTGF) and B-cell lymphoma-2 (Bcl-2) and decrease expression of Bcl-2 associated X protein (Bax).	[32]
**Proliferation** **(Cell cycle and growth)**	Cyclin and CDK	YAP, as well as mutant p53 and the transcription factor NF-Y, bind onto the regulatory regions of mutant p53 pro-proliferative transcriptional activity genes, such as cyclin A, cyclin B, and CDK1.	[33]
	MCMs	Hyperactivated YAP in gastric cancer (GC) induces MCM transcription via binding to its promoter. The YAP–MCM axis facilitates GC progression by inducing PI3K/Akt signaling.	[34]
	CDC25	CDC25 is a member of the phosphatase family and is a protein phosphatase that plays an important role in the regulation of the cell cycle. Activation of YAP/TAZ/YKI may lead to the upregulation of CDC25/string.	[35]
	SMAD	SMADs activated by TGF-β translocate into the nucleus and bind to YAP, thus promoting the expression of the target gene and cell proliferation.	[13]
	TERT	The Hippo pathway effector Yes-associated protein (YAP) promotes the expression of human telomerase reverse transcriptase (hTERT). YAP transcriptionally activates the hTERT promoter through interaction with TEAD.	[36]
**Angiogenesis**	VEGF	YAP/TAZ activity is controlled by VEGF during angiogenesis. VEGF induces a YAP/TAZ-dependent transcriptome linked to cytoskeleton remodeling.	[37]
	Axl	YAP/TAZ promote angiogenesis by fueling nutrient-dependent mTORC1 signaling. The upregulated genes observed were prototypical YAP/TAZ targets, such as CTGF, AXL, CYR61, as well as numerous genes linked to mechanistic targeting of mTORC1 signaling.	[38]
	CTGF	YAP/TAZ are activated by blood circulation in the endothelial cells. This leads to induction of CTGF and actin polymerization.	[39]
	Ang2	Overexpression of an active form of YAP promotes hypersprouting via angiogenic growth factor angiopoietin-2 (Ang2) signaling. Hypoxia stabilizes hypoxia inducible transcription factor 1α (HIF1α) in tumor cells, initiating the transcription and secretion of pro-angiogenic factors such as VEGF and Ang2.	[23,37]
	MMPs	YAP/TAZ-mediated tumor angiogenesis occurs through MMP-mediated ECM remodeling.	[23]
	MCMs	YAP/TAZ promote EC proliferation in a MCM-dependent manner.	[40]
**Immune Suppression**	CXCL5	YAP, in complex with TEAD in cancer cells, stimulates the recruitment of MDSCs within the TME by transcriptionally inducing the production of cytokines, including CSF and CXCL5.	[41]
	CCL2	YAP and TAZ bind to the Ccl2 promoter. Increased TAZ expression was correlated with increased expression of the inflammatory cytokine CCL2.	[41,42]
	TGF-beta	The expression of YAP is increased in Tregs, and signaling through YAP increases SMAD/TGFβ signaling and promotes Treg differentiation.	[43]
	IL-10	Activation of the YAP/TAZ–TEAD pathway increases the proportion of MDSCs and enhances the expression of the immunosuppressive cytokines IL-10 and INF-r, which promote Treg cell proliferation.	[44]
	PD-L1	TAZ promotes immune evasion in human cancer through PD-L1; TAZ/YAP/TEAD increase PD-L1 promoter activity. TAZ-induced PD-L1 upregulation in human cancer cells is sufficient to inhibit T-cell function	[45,46]
	IL-35	Tregs in YAP-induced TME secrete the cytokines TGF-β, interleukin-10 (IL-10), and interleukin-35 (IL-35) to maintain their immunosuppressive effects.	[47]
	ICOS	When YAP-induced, high expression of TEAD4 regulates immune checkpoint genes (PDCD1, IDO1, ICOS), cytokines (IL-10, CXCL11), cytokine receptors (CCR2, CXCR3, CXCR6, IL2RA), and some other mediators of immune function.	[48]

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
