# Peer review of "Regulation of Tumor Microenvironment through YAP/TAZ under Tumor Hypoxia"

_cancers, 2024, doi:10.3390/cancers16173030_

Round 1

Reviewer 1 Report

Comments and Suggestions for Authors

In this work, Choi and Kim review the importance of Hippo effectors YAP/TAZ in the hypoxic tumor microenvironment with a focus on liver cancers. Overall, the review is of significance and should be of interest to the YAP/TAZ community. However, several aspects of the regulation of YAP/TAZ on the TME about angiogenesis and hypoxia creation are not elucidated in the current version. Therefore., I would recommend the authors stress several of the points noted below to improve this review's readability and broad impact.

Specific comments

1) The authors have not discussed the specific TME compartments and how YAP/TAZ regulates hypoxia in these compartments. For instance, they focus on cancer cells (preventing apoptosis, cell cycle progression on page 2), but do not discuss how this differs in endothelial cells versus cancer cells versus other cell types of the TME. 

2) The role of the extracellular matrix in regulating endothelial cell migration and the creation of hypoxia needs to be added as this represents one of the major factors of time. Several pieces of work needs to be referred (for instance: PMID: 36797231, those in HCC domains: PMID: 38956074; PMID: 37652015)

3) Fig 1: needs revision to incorporate the mechanical signals from the MATRIX to YAP/TAZ. Moreover, the legend is not correctly reflecting the physiological roles of YAP and TAZ. They are paralogues and do not necessarily interact or separate out during hypoxia. Each of them (YAP and TAZ) have overlapping and/or redundant roles. This needs to be clearly articulated.

4) The role of YAP/TAZ in regulating immunotherapy response linked to Hypoxia needs to be discussed as done in PMID: 37444578; PMID: 38550587

5) Page 7 and Fig 3: What about phosphorylation-independent roles of YAP and TAZ through actin remodeling and or other Hippo cascade? This also should be briefly discussed

volume8, Article number: 70 (2023

Comments on the Quality of English Language

minor editing recommended

Author Response

Reviewer 1 Comment

In this work, Choi and Kim review the importance of Hippo effectors YAP/TAZ in the hypoxic tumor microenvironment with a focus on liver cancers. Overall, the review is of significance and should be of interest to the YAP/TAZ community. However, several aspects of the regulation of YAP/TAZ on the TME about angiogenesis and hypoxia creation are not elucidated in the current version. Therefore., I would recommend the authors stress several of the points noted below to improve this review's readability and broad impact.

Specific comments

1) The authors have not discussed the specific TME compartments and how YAP/TAZ regulates hypoxia in these compartments. For instance, they focus on cancer cells (preventing apoptosis, cell cycle progression on page 2), but do not discuss how this differs in endothelial cells versus cancer cells versus other cell types of the TME. 

Answer: Thanks for your comments, we added an angiogenesis part to the text to explain TME as endothelial cells.

2) The role of the extracellular matrix in regulating endothelial cell migration and the creation of hypoxia needs to be added as this represents one of the major factors of time. Several pieces of work needs to be referred (for instance: PMID: 36797231, those in HCC domains: PMID: 38956074; PMID: 37652015)

Answer : We added the angiogenesis part to the body and explained the metastasis as well

3) Fig 1: needs revision to incorporate the mechanical signals from the MATRIX to YAP/TAZ. Moreover, the legend is not correctly reflecting the physiological roles of YAP and TAZ. They are paralogues and do not necessarily interact or separate out during hypoxia. Each of them (YAP and TAZ) have overlapping and/or redundant roles. This needs to be clearly articulated.

Answer: We've added a more detailed explanation of the illustration in response to your review.

4) The role of YAP/TAZ in regulating immunotherapy response linked to Hypoxia needs to be discussed as done in PMID: 37444578; PMID: 38550587

The section on the role of YAP/TAZ in regulating hypoxia-associated immunotherapy responses was difficult to add because the content on the effects of YAP/TAZ on immune cells and treatment was too extensive. In addition, the content of the text was focused on the cancer biological functions of YAP/TAZ in hypoxic environments, so it was difficult to insert the content on treatment, so we will organise and present the relevant content at a later date if there is an opportunity. We apologise for the omission. 

5) Page 7 and Fig 3: What about phosphorylation-independent roles of YAP and TAZ through actin remodeling and or other Hippo cascade? This also should be briefly discussed

Answer: We added content on phosphorylation independent signals such as Actin remodeling.

Reviewer 2 Report

Comments and Suggestions for Authors

In this paper, Choi SH and Kim DY et al., have reviewed the Regulation of tumor microenvironment through the YAP/TAZ under tumor hypoxia.

Here is the summary of the review.

Hypoxia significantly influences hepatocellular carcinoma (HCC) development, survival, and metastasis. Although immunotherapy has advanced HCC treatment, it faces challenges like limited response rates, resistance, and high costs. HCC, being a highly vascular tumor, is particularly sensitive to hypoxia, complicating treatment with immunotherapy and anti-vascular drugs. Understanding and targeting hypoxic mechanisms are crucial for developing effective HCC treatments. The tumor growth mechanisms linked to hypoxia are complex, involving various signaling pathways, including the emerging Hippo signaling pathway. The Hippo YAP/TAZ proteins, which play roles in both early and hypoxic tumor growth, are critical for tumor proliferation. Understanding how YAP/TAZ functions and its regulation in hypoxic conditions, along with HIF, can lead to new anti-cancer strategies. Combining HIF and YAP/TAZ targeting may offer promising new approaches in cancer treatment.

This review paper is interesting and contains certain new aspects for the potential therapeutics.

I have several comments to further clarify the content and makes more informative.

1.     Several sentences are redundant and repetitive.

For example,

“Regulation of DNA repair pathways YAP/TAZ, as transcriptional coactivaors, can modulate the expression of genes involved in DNA repair pathways[13]. Under hypoxic conditions, YAP/TAZ activation promotes the transcription of genes involved in DNA repair mechanisms[14]”

“YAP can activate the expression of genes involved in promoting cell survival, proliferation, and angiogenesis, contributing to tumor growth and progression under hypoxia. These target genes may include those encoding for angiogenic factors, glycolytic enzymes, and anti-apoptotic proteins.”

This kind of description appears repetitively. Please review the entire manuscript.

2.     “They may exhibit differences in their regulatory mechanisms, transcriptional targets, cellular localization, and functional roles (Figure 1)[31, 32]”

This can’t be reflected in the Figure 1.

3.     What do GD1a, GD2,,,,GM3 mean in Figure 2?

4.     Although the title of Figure 2 includes YAP/TAZ, it is unclear that Figure 2 contains information regarding YAP. They seem to be all related to TAZ, don’t they? Unless otherwise please specify.

5.     YAP and TAZ phosphorylation seems to be different but it was not well reflected in Figure 3.

Comments on the Quality of English Language

Some sentences use wrong grammar and English editing is recommended 

Author Response

Reviewer 2 Comment

In this paper, Choi SH and Kim DY et al., have reviewed the Regulation of tumor microenvironment through the YAP/TAZ under tumor hypoxia.

Here is the summary of the review.

Hypoxia significantly influences hepatocellular carcinoma (HCC) development, survival, and metastasis. Although immunotherapy has advanced HCC treatment, it faces challenges like limited response rates, resistance, and high costs. HCC, being a highly vascular tumor, is particularly sensitive to hypoxia, complicating treatment with immunotherapy and anti-vascular drugs. Understanding and targeting hypoxic mechanisms are crucial for developing effective HCC treatments. The tumor growth mechanisms linked to hypoxia are complex, involving various signaling pathways, including the emerging Hippo signaling pathway. The Hippo YAP/TAZ proteins, which play roles in both early and hypoxic tumor growth, are critical for tumor proliferation. Understanding how YAP/TAZ functions and its regulation in hypoxic conditions, along with HIF, can lead to new anti-cancer strategies. Combining HIF and YAP/TAZ targeting may offer promising new approaches in cancer treatment.

This review paper is interesting and contains certain new aspects for the potential therapeutics.

I have several comments to further clarify the content and makes more informative.

1.Several sentences are redundant and repetitive.For example,

“Regulation of DNA repair pathways YAP/TAZ, as transcriptional coactivaors, can modulate the expression of genes involved in DNA repair pathways[13]. Under hypoxic conditions, YAP/TAZ activation promotes the transcription of genes involved in DNA repair mechanisms[14]”

“YAP can activate the expression of genes involved in promoting cell survival, proliferation, and angiogenesis, contributing to tumor growth and progression under hypoxia. These target genes may include those encoding for angiogenic factors, glycolytic enzymes, and anti-apoptotic proteins.”

This kind of description appears repetitively. Please review the entire manuscript.

Answer : We have fixed the overlap you indicated

2.They may exhibit differences in their regulatory mechanisms, transcriptional targets, cellular localization, and functional roles (Figure 1)[31, 32]”

This can’t be reflected in the Figure 1.

Answer: We've added a more detailed explanation of the illustration in response to your review.

3. What do GD1a, GD2, GM3 mean in Figure 2?

Answer : The terms Gd1, GD2, etc. have been removed from the figure as they do not fit with the overall content.

4. Although the title of Figure 2 includes YAP/TAZ, it is unclear that Figure 2 contains information regarding YAP. They seem to be all related to TAZ, don’t they? Unless otherwise please specify.

Answer : In the figure and in the text, we have written more about YAP.

5.YAP and TAZ phosphorylation seems to be different but it was not well reflected in Figure 3.

Answer: We've added more explanations and modified the figure3 to reflect your feedback

Reviewer 3 Report

Comments and Suggestions for Authors

In the current study, the author reviewed the recent progress of YAP/TAZ in hypoxia tumor microenvironment. In particular, the mechanisms of the interplay of YAP/TAZ and HIF in regulating tumorigenesis. Overall, this review study is enlightening and instructive and, to some extent, will benefit the field in the future. My specific comments are listed below.

1.      The title “Regulation of tumor microenvironment through the YAP/TAZ 2 under tumor hypoxia”. However, in the “Abstract” and “Introduction”, the authors mainly discussed hepatocellular carcinoma. In the main text, it’s kind of logical mess when I read the paper, especially the parts related to these cancer types. Please revise the manuscript in a more logical way.

2.      In all 3 Figures, every molecular/protein/gene shown in the diagram should be mentioned, either in the main text or in the figure legends, or both. Otherwise, why do you show them to the readers? For example, in Figure 1, what’s the role of PKM2 here? In figure 2, too much information shown in the diagram was not mentioned or discussed.

Author Response

Reviewer 3 comments

In the current study, the author reviewed the recent progress of YAP/TAZ in hypoxia tumor microenvironment. In particular, the mechanisms of the interplay of YAP/TAZ and HIF in regulating tumorigenesis. Overall, this review study is enlightening and instructive and, to some extent, will benefit the field in the future. My specific comments are listed below.

1. The title “Regulation of tumor microenvironment through the YAP/TAZ 2 under tumor hypoxia”. However, in the “Abstract” and “Introduction”, the authors mainly discussed hepatocellular carcinoma. In the main text, it’s kind of logical mess when I read the paper, especially the parts related to these cancer types. Please revise the manuscript in a more logical way.

Answer: As you mentioned, we've changed the focus to solid cancers rather than HCC. HCC was described as an example of a solid cancer.

2. In all 3 Figures, every molecular/protein/gene shown in the diagram should be mentioned, either in the main text or in the figure legends, or both. Otherwise, why do you show them to the readers? For example, in Figure 1, what’s the role of PKM2 here? In figure 2, too much information shown in the diagram was not mentioned or discussed.

Answer: We've added a more detailed explanation of the illustration in response to your review.

Round 2

Reviewer 1 Report

Comments and Suggestions for Authors

The authors have by most parts addressed my previous concerns. Therefore, the the review is more updated and comprehensive for publication.

Comments on the Quality of English Language

no issues on language

Reviewer 3 Report

Comments and Suggestions for Authors

The authors revised their manuscript and provided relatively reasonable explanations for my questions. Some details were also added into this manuscript. This review and discussion are integrated and more convincing. This manuscript should be able to bring the readers some understanding about the role of Hypoxia-Inducible Factor and YAP/TAZ in tumor microenvironment. I have no more questions for the current version.